# Carbonised Human Hair Incorporated in Agar/KGM Bioscaffold for Tissue Engineering Application: Fabrication and Characterisation

**DOI:** 10.3390/polym14245489

**Published:** 2022-12-15

**Authors:** Vieralynda Vitus, Fatimah Ibrahim, Shamsul Azlin Ahmad Shamsuddin, Nuguelis Razali, Noor Anastasha Balqis Noor Azlan, Wan Safwani Wan Kamarul Zaman

**Affiliations:** 1Department of Biomedical Engineering, Faculty of Engineering, Universiti Malaya, Kuala Lumpur 50603, Malaysia; 2Centre for Innovation in Medical Engineering (CIME), Department of Biomedical Engineering, Faculty of Engineering, Universiti Malaya, Kuala Lumpur 50603, Malaysia; 3Centre for Printable Electronics, Universiti Malaya, Kuala Lumpur 50603, Malaysia; 4Institute of Biological Science, Faculty of Science, Universiti Malaya, Kuala Lumpur 50603, Malaysia; 5Department of Obstetrics and Gynaecology, Faculty of Medicine, Universiti Malaya, Kuala Lumpur 50603, Malaysia

**Keywords:** human hair, carbon, pyrolysis, biocompatibility, Wharton’s jelly-derived mesenchymal stem cells

## Abstract

Carbon derived from biomass waste usage is rising in various fields of application due to its availability, cost-effectiveness, and sustainability, but it remains limited in tissue engineering applications. Carbon derived from human hair waste was selected to fabricate a carbon-based bioscaffold (CHAK) due to its ease of collection and inexpensive synthesis procedure. The CHAK was fabricated via gelation, rapid freezing, and ethanol immersion and characterised based on their morphology, porosity, Fourier transforms infrared (FTIR), tensile strength, swelling ability, degradability, electrical conductivity, and biocompatibility using Wharton’s jelly-derived mesenchymal stem cells (WJMSCs). The addition of carbon reduced the porosity of the bioscaffold. Via FTIR analysis, the combination of carbon, agar, and KGM was compatible. Among the CHAK, the 3HC bioscaffold displayed the highest tensile strength (62.35 ± 29.12 kPa). The CHAK also showed excellent swelling and water uptake capability. All bioscaffolds demonstrated a slow degradability rate (<50%) after 28 days of incubation, while the electrical conductivity analysis showed that the 3AHC bioscaffold had the highest conductivity compared to other CHAK bioscaffolds. Our findings also showed that the CHAK bioscaffolds were biocompatible with WJMSCs. These findings showed that the CHAK bioscaffolds have potential as bioscaffolds for tissue engineering applications.

## 1. Introduction

The element carbon is widely distributed across the planet in a variety of allotropes [1]. In terms of physicochemical characteristics, carbon is superior to other materials such as metal and silicon [2]. Additionally, carbon is one of the most prevalent elements that our body uses as a building block, primarily in the form of hydrocarbon chains found in DNA components, lipids, proteins, carbohydrates, etc. [1,3]. As a result, carbon is being used more frequently as a biomaterial, especially in the creation of bioscaffolds or other technology for biomedical and tissue engineering purposes. In addition, the addition of carbon materials into the bioscaffold improves the bioscaffold’s mechanical strength and biocompatibility with different cell lineages [4,5]. For example, a carbon nanofiber-incorporated collagen bioscaffold reportedly could promote neovascularisation and regeneration of cardiac tissue in a myocardial ischemia/reperfusion model in rats [6]. Another study showed that polyethylene glycol (PEG)-functionalized carbon nanotubes (CNT) incorporated into the silk fibroin (PEG-CNT/SF) bioscaffold could photoacoustically stimulate neural cells and promote neurite outgrowth [7]. Moreover, electrospun nanofibrous bioscaffolds incorporated with CNT are biocompatible with H9C2 cells [8]. Nonetheless, conventional synthesis methods and relatively expensive raw materials or non-renewable (e.g., coal, lignite, peat, SU-8 photoresist) materials used to synthesise carbon products have led to the high cost of production, which may affect their commercialisation potential [9,10,11]. Hence, this has encouraged researchers and manufacturers to find an alternative that is cost-effective and sustainable.

Considering these factors, the use of biomass wastes as raw materials to synthesise carbon products is rising [12]. Natural biomass wastes provide a better source of carbon precursors, as they are abundant and easily accessible. In addition, the use of biomass wastes has allowed for sound management of its accumulation, which could potentially be harmful to human well-being and the environment. Various biomass waste sources are currently being researched as potential carbon precursor materials [13,14,15,16,17]. Among all of these potential biomass wastes, human hair provides an inexpensive source of materials, and its usage could prevent zoonotic transmission, which can be deadly to humans. Moreover, humans shed approximately 50 to 150 strains of hair per day, and hair is one of the wastes commonly produced by human hair-based industries. This could lead to the accumulation of human hair that can negatively affect human health and well-being, and the environment. Human hair waste can potentially cause blockage in the sewage system when tufts of hair tangle in the system. Furthermore, tufts of hair could potentially be a matrix for dust to accumulate and can become a microorganism habitat in wet states. Therefore, hair waste management is important. The effort to tackle human hair waste accumulation is ongoing, and one of the strategies is synthesising carbon from human hair and using it as material to develop a functional product. To date, carbon derived from human hair has shown its potential application as electrochemical sensors [18], probes [19], and supercapacitors [20]. Despite that, the usage of carbonised human hair in bioscaffold development for tissue engineering applications is limited.

Furthermore, carbon material alone is insufficient to fabricate a bioscaffold suitable for tissue engineering applications. Therefore, combining it with other biomaterials such as natural polymers could improve the functionality of the bioscaffold, which is more suitable for tissue engineering applications. Among the existing natural polymers, konjac glucomannan (KGM) application in tissue engineering, especially in bioscaffold fabrication, is growing. KGM is a natural polysaccharide extracted from *Amorphophallus konjac* tubers [21]. KGM is largely made up of linear polymer consisting of D-glucose and D-mannose linked by β-D-1,4 bonds [22]. It is biodegradable, biocompatible, hydrophilic, and physicochemically stable, and it has film-forming properties [22,23]. Due to its beneficial properties, it has been widely applied in various fields including food industries [24,25], food packaging [26], medicine [27,28], drug carriers [29,30], and tissue engineering [31,32]. However, there is a limitation to using pure KGM film, mainly because of its hydrophilicity and weak mechanical strength [21]. These drawbacks can be overcome by blending with another polymer. Reinforcement strategies of KGM film include the blending of gum Arabic [33], zein [34], agar [35], and welan gum [36]. Agar is a polysaccharide derived from red algae, which is composed of 1,3-linked β-D-galactopyranose and 1,4-linked 3,6-anhydro-α-L-galactopyranose or agarobiose [37]. It is a renewable and easily accessible biopolymer, which has been used as gelling, thickening, water-holding, and stabilising agents in various fields including edible food and food packaging, biotechnology industries, medical fields, and pharmaceuticals [35,38,39,40]. Previous studies have shown positive effects of blending agar with KGM on the mechanical properties of film such as tensile strength, flexibility, and hydrophilicity [37,41].

Additionally, the size of the global scaffold technology market was estimated at USD 1.1 billion in 2020, and it is anticipated to increase at a compound annual growth rate of 8.4% from 2021 to 2028 [42]. The increasing need for three-dimensional (3D) cellular models in biological studies and translational research is primarily responsible for the market’s expansion. The market for scaffold technology is predicted to expand because of the quick paradigm shift in cell culture that is needed to address the difficulties in the drug development process. The use of 3D cell cultures is expanding due to their effectiveness in simulating the physiological conditions found in vivo for the accurate representation of disease-causing microenvironmental variables. As a result, 3D cell culture techniques have become a cutting-edge tool for the early discovery of drugs as well as possible therapeutic options for the treatment of a range of disorders. By giving pertinent information about diseased cells, the 3D idea of cell cultivation enables the analysis of the phenotypic heterogeneity of cancer cells and heterotypic communication between cells, thus propelling the market [42]. Therefore, the investigation of a new bioscaffold as a cellular model with different materials or incorporating newer materials, such as carbon, to an existing developed bioscaffold is a relevant study to produce effective therapeutic tools with commercialisation potential.

The beneficial use of carbon materials in facilitating cell or tissue growth has influenced its application in tissue engineering by combining it with other biomaterials to develop a bioscaffold that has the same biological characteristics as the natural extracellular matrix (ECM) of the human body. In this regard, inexpensive and renewable carbon sources as well as facile synthesis methods to produce carbon from biomass waste (e.g., human hair) have offered non-costly carbon products. Furthermore, to the best of our knowledge, information concerning the effects of carbon derived from human hair incorporated in agar/KGM bioscaffolds and their application for cell development has not been reported. Therefore, developing this bioscaffold could help to better understand the feasibility of using carbon derived from biomass as an alternative to existing expensive carbon such as graphite. Hence, the objective of the present study was to develop a CHAK based on agar and KGM incorporated with carbon synthesised from human hair. The structural, physical, and biocompatibility of the bioscaffold were investigated. The results of this work show that human hair-derived carbon can be used to make bioscaffolds for tissue engineering applications.

## 2. Materials and Methods

### 2.1. Human Hair-Derived Carbon

Human hair waste treated with bleaching and colouring was obtained from a single female donor. The collected hair sample was rinsed with a 70% ethanol solution and distilled water to remove any adhered debris, such as dandruff and dust, and then was dried in an oven at 50 °C for 2 h. In this study, two types of carbon that are non-activated and activated carbon were synthesised.

#### 2.1.1. Synthesis of Non-Activated Carbon

The synthesis of non-activated carbon was carried out by pyrolysing the hair sample at 900 °C in an open-ended quartz tube furnace (Nabertherm, Lilienthal, Germany) with a constant nitrogen gas flow at 2000 mL/min. Then, the pyrolysed carbon was pulverised into fine micro-powder and kept in a sealed jar for further characterisation and use. The non-activated carbon obtained was then referred to as HC.

#### 2.1.2. Synthesis of Activated Carbon

The activated carbon synthesis was carried out by pyrolysing the hair sample at 300 °C for 1 h 30 min in the furnace with a constant nitrogen gas flow at 2000 mL/min. The cooled char was then mixed with a 1 M potassium hydroxide (KOH) solution in a 3:1 ratio. The mixture was then pyrolysed for 1 h at 800 °C in the furnace in a nitrogen environment. After that, the cooled carbon was mixed with 100 mL of 1 M HCl in a beaker and stirred for 2 h. The mixture was then filtered and rinsed with PBS (pH 7.4) until the neutral pH of the filtrate was obtained. The filtered precipitate was placed in an oven at 90 °C until it was dry. Finally, the dried carbon was pulverised into fine micro-powder and kept in a sealed jar for further characterisation and use. The activated carbon obtained was then referred to as AHC.

#### 2.1.3. Characterisation of Carbon

Morphological characterisation of raw human hair, carbonised human hair before pulverisation, HC, and AHC were observed and analysed using tabletop scanning electron microscopy (SEM, Phenom ParticleX Steel Desktop SEM, Thermo Scientific, Waltham, MA, USA). HC and AHC were analysed using an X-ray diffractometer (XRD, Empyrean, Panalytical, Worcestershire, United Kingdom) with a Cu-Kα1 source. The X-ray generator was 40 kV/40 mA, and the detector was a two-dimensional semiconductor X-ray detector. XRD diffraction peaks were obtained in 2θ continuous scanning mode within a scanning range of 0–60° and at a scanning speed of 1° per min.

### 2.2. Fabrication of Bioscaffold

This study fabricated six types of bioscaffolds, including pure agar (A), agar/KGM (AK), 0.01 g HC/agar/KGM (1HC), 0.03 g HC/agar/KGM (3HC), 0.01 g AHC/agar/KGM (1AHC), and 0.03 g AHC/agar/KGM (3AHC).

The A bioscaffold was fabricated with agar powder (0.1 g) mixed in phosphate-buffered saline (PBS, pH 7.4, 10 mL) and heated to 98 °C on the hot plate. Moreover, the AK bioscaffold was fabricated with agar (0.1 g) and KGM (0.1 g) powder mixed in PBS (10 mL) and heated to 98 °C. Similarly, the CHAK was fabricated by mixing the composite materials consisting of agar and KGM and carbon in PBS (10 mL) and heated to 98 °C (Figure 1). Then, the resulting molten solution was slightly cooled and poured into the mould to set, which was prepared using poly (methyl methacrylate) (PMMA) glass. Three PMMA glasses were aligned at the bottom, and then two PMMA glasses with a 15 mm distance between each other were stacked on top of the three PMMA glasses. Then, the molten mixture was poured into the mould and covered with PMMA glass. The thickness of the PMMA glass was 4 mm. After cooling, the PMMA glass was removed, and the gel was cut at 1.5 cm × 1.0 cm, followed by freezing for 2 h at –20 °C. Then it was thawed at room temperature and immersed in 95% ethanol for 1 h with gentle stirring several times. Lastly, it was allowed to dry in the fume hood.

### 2.3. Characterisation of Bioscaffold

#### 2.3.1. Morphological Analysis

Morphological characterisation of the bioscaffold was observed using scanning electron microscopy (SEM, Phenom ParticleX Steel Desktop SEM, Thermo Scientific, Waltham, MA, USA). The porosity of the dried bioscaffold was determined from the SEM results using ImageJ analysis software (Java 1.8.0_172 (64-bit)).

#### 2.3.2. FTIR Analysis

FTIR analysis was performed to determine the possible interaction between carbon and the composite materials. The FTIR spectrum of agar, agar and KGM, and CHAK were analysed using a spectrometer (Nicolet™ Summit PRO, Spectrometer, Thermo Scientific, Waltham, MA, USA), whereby the bioscaffolds were analysed in the dried state. The analysis was performed at a wavelength range of 4000–500 cm^−1^ at a resolution of 2 cm^−^¹ per point at room temperature and evaluated for the carbon and excipients interaction by analysing characteristic peaks.

#### 2.3.3. Swelling and Water Uptake Analysis

In this test, the bioscaffold’s dry weight (Wi) was first measured, and then the bioscaffold was immersed in PBS for 1 h at 37 °C. The excess water on the bioscaffold was removed by gentle dabbing with filter paper, and the swollen bioscaffold weight (Wf) was measured. The swelling capacity of the bioscaffold was calculated as the ratio of PBS absorbed to the initial dry weight of the scaffold, whereas the water uptake capability of the bioscaffolds was measured as the ratio of absorbed PBS to the swollen weight of the bioscaffolds. Measurements were made in triplicate.
(1)$$\mathrm{Degree}\;\mathrm{of}\;\mathrm{Swelling}\;\left(\%\right)=\frac{\left(\mathrm{Wf}\;-\;\mathrm{Wi}\right)}{\mathrm{Wi}}\times 100\%$$
(2)$$\mathrm{Water}\;\mathrm{Uptake}\;\mathrm{Capacity}\;\left(\%\right)=\frac{\left(\mathrm{Wf}\;-\;\mathrm{Wi}\right)}{\mathrm{Wf}}\times 100\%$$

#### 2.3.4. Degradation Analysis

Initially, the weight of the bioscaffold was measured, and then the degradation analysis of the bioscaffold was carried out under static conditions for up to 28 days immersed in PBS (pH 7.4) at room temperature. The PBS was changed after 14 days of incubation for 21 days and 28 days of the group study. On analysis day, the excess PBS was removed by gently dabbing the bioscaffold on filter paper and placed it in the oven at 50 °C until completely dry. Then, the final weight of the dried bioscaffold was measured. Measurements were made in triplicate.

#### 2.3.5. Tensile Test

The tensile test was performed using a universal testing machine (AG-X Series, SHIMADZU, Kyoto, Japan). Both ends of the bioscaffold were clamped and sprayed with phosphate buffer saline (PBS, pH 7.4) to rehydrate the bioscaffold. Then, the bioscaffold was retracted with a sample length of 10 mm and a speed of 2 mm/min. The measurement of the tensile strength was carried out on the semi-wet state of the bioscaffold.

#### 2.3.6. Electrical Conductivity Analysis

Initially, the bioscaffold was immersed in PBS for 1 h. Then, two 70% alcohol-cleaned steel blades were placed on a non-conductive platform at a 0.35 cm distance from one another. Electrical conductivity was carried out by placing the hydrated bioscaffold between the steel blades, and the resistivity of the bioscaffold characterization was measured using a digital multimetre. The electrical conductivity was evaluated by its resistance activity. Five resistance (R) measurements were performed on each sample, and the associated conductivity (δ) of the bioscaffold was calculated using the Pouillet’s law equation:(3)$$\delta =\frac{L}{R\times A}$$
where R represents the resistance of the hydrated bioscaffold, L indicates the length, and A represents the surface area of samples, $A=\;\pi \times {\left(D/2\right)}^{2}$, while D is the diameter of the bioscaffold [43].

### 2.4. Cell Culture

#### 2.4.1. Preparation of WJMSCs

This study was approved by the Medical Research Ethics Committee of the University of Malaya Medical Centre (MREC ID No: 2021518-10145). Wharton’s jelly-derived mesenchymal stem cells (WJMSCs) isolated from the human umbilical cord were used to conduct biocompatibility analysis on the fabricated bioscaffold. Initially, the collected umbilical cord was immersed in 70% ethanol for 30 s followed by rinsing twice with PBS. Then, the inner lining of the umbilical cord tissue, known as Wharton’s jelly, was cut into approximately 5 mm in diameter and rinsed with PBS to remove the blood clots. Next, the tissue was transferred into a T25 flask containing complete media comprising 90% DMEM/F-12 (ATCC), 10% Gibco Foetal Bovine Serum (FBS; Thermo Fisher Scientific, Waltham, MA, USA, Europe approved, South American Origin), 1% of Gibco antibiotic–antimycotic solution (100×, Thermo Fisher Scientific), and 1% of Gibco Glutamax supplement (Thermo Fisher Scientific) and incubated at 37 °C in a humid atmosphere of 5% CO_2_. The isolated cells were trypsinised and grown until passage 3.

#### 2.4.2. Characterisation of WJMSCs

Characterisation of the WJMSCs was performed based on their morphology, surface marker, and differentiation ability. Immunophenotyping analysis to detect the surface marker of the WJMSC was conducted according to the manufacturer protocol using a BD Stemflow Human MSC Analysis Kit (BD Biosciences, San Jose, CA, USA). The differentiation ability of WJMSCs was performed using the Human Mesenchymal Stem Cell Functional Identification Kit (R&D Systems, Minneapolis, MN, USA) according to the manufacturer’s protocol.

#### 2.4.3. Biocompatibility Analysis of Bioscaffold

Before the biocompatibility study, the bioscaffolds were placed in a 24-well tissue culture plate and immersed in 70% ethanol, then left to dry in a biosafety cabinet under UV light for at least 2 h. Then the bioscaffolds were rinsed twice with PBS for 10 min each. Subsequently, the bioscaffolds were pre-soaked in 0.5 mL of complete medium containing 90% DMEM/F-12, 10% of FBS, 1% of the antibiotic–antimycotic solution, and 1% of glutamax supplement for 30 min. Then, a concentration of 3 × 10^4^ cells per well of WJMSCs was cultured on the bioscaffold. The bioscaffold was then incubated at 37 °C in a humid atmosphere of 5% CO_2_.

Biocompatibility analysis was measured using the MTT (3-(4,5-Dimethylthiazol-2-yl)-2,5-Diphenyltetrazolium Bromide) assay. The biocompatibility analysis was performed at 1 day, 3 days, and 7 days of incubation. After the period of incubation, the conditioned medium was discarded and washed with PBS, followed by the detachment of cells using trypsin-EDTA (trypsin-ethylenediamine tetraacetic acid) solution. Then, 1000 µL of complete media was added to stop the trypsin-EDTA activity. The detached cells were transferred to a centrifuge tube and centrifuged at 2000 rpm for 1 min. The cells were resuspended with 500 µL of complete media and incubated in a 24-well tissue culture plate for 45 min at 37 °C in a humid 5% CO₂ atmosphere. After incubation, the complete medium was replaced with 500 µL of a complete media containing MTT reagent (0.5 mg/mL). The cells were then incubated for 4 h at 37 °C in the dark. After 4 h, the working solution was removed, and 500 µL of dimethyl sulfoxide (DMSO; Sigma-Aldrich, Saint Louis, MO, USA) was added to dilute the formazan crystal product for 18 min. The production of the formazan crystal absorbance value was then measured by using a microplate reader at a 570 nm wavelength. The observations and measurements were performed in triplicate.

### 2.5. Statistical Analysis

Statistical analysis was conducted using one-way and two-way analyses of variance (ANOVA), followed by a Tukey post hoc test for multiple comparisons to confirm significant differences among samples using GraphPad Prism software (Dotmatics, San Diego, CA, USA). The data from this study were obtained in triplicate for each group, except for porosity and electrical conductivity analysis data, which were obtained in quintuplet, and the data were presented as mean value ± standard deviation (SD). The statistical significance was considered at a probability of *p* < 0.05 (* *p* ≤ 0.05, ** *p* ≤ 0.01, *** *p* ≤ 0.001, **** *p* ≤ 0.0001).

## 3. Results and Discussion

### 3.1. Carbon-Derived Human Hair

Following the synthesis of carbonised human hair, SEM micrographs of human hair and carbon samples were analysed and are presented in Figure 2. Human hair is comprised of 3 main layers, which are the outer cuticle layer, the inner cortex layer, and the innermost layer of the medulla (Figure 2A) [44]. The SEM micrograph of raw human hair showed the outer cuticle layer surrounding the hair fibre (Figure 2D), and the cross-sectional micrograph of raw hair (Figure 2B) showed the three main layers that make up human hair. After pyrolysis, the resulting carbonised human hair showed a hollow microfibre structure similar to the result obtained by Pramanick et al. (2016) (Figure 2C) [45]. The medulla layer’s removal and the fusion of the cuticle and cortical layers led to the formation of a hollow carbon microfibre [45].

In this study, two forms of carbon powder derived from human hair were synthesised using the pyrolysis process: (i) non-activated human hair-derived carbon (HC), and (ii) activated human hair-derived carbon (AHC). Both pulverised carbons exhibited microparticle sizes (<200 µm). The synthesised HC had a rough surface without porous features (Figure 2E), whereas the AHC showed a porous surface (Figure 2F). The resulting porous structure was due to the KOH chemical treatment. The KOH helped to remove volatile substances during activation [46]. The following reaction is a representation of the creation of porosity by KOH activation [47]:$$6\mathrm{KOH}+2C\;\to \;2K+3H_{2}+2K_{2}CO_{3}$$

Additionally, potassium carbonate (K_2_CO_3)_ reaction with carbons created potassium (K), potassium oxide (K_2_O), carbon monoxide (CO), and carbon dioxide (CO_2)_, which increased the creation of pores. It was hypothesised that during the carbonisation process, KOH was converted to metallic potassium K, which entered the carbon particle and created the pores [47]. A study reported similar findings on KOH-activated pollen grains [48]. The porous carbon structure produced by the activation of carbon with an activating chemical at high temperatures resembles characteristics identified by other studies [46,49,50,51].

Additionally, the degree of crystallinity of the synthesised HC and AHC was evaluated through XRD analysis, and the XRD pattern is shown in Figure 3. The relative fractions of the crystalline were evaluated based on the maximum intensity of the 002 peaks. The XRD patterns had a very high baseline and a wide peak range. The diffraction angle (2θ) of the 002 peaks of crystalline carbon was around 24.03° and 22.24° for AHC and HC, respectively, whereas the human hair had 002 peaks located at 20.19°. This indicated that the micro-crystallites of hair shifted after carbonisation. Moreover, there were slight changes in the 002 peak between HC and AHC, indicating the activation process caused a shift of the micro-crystallites, thus resulting in a porous carbon structure.

### 3.2. Fabrication and Characterisation of Carbon-Based Bioscaffold

#### 3.2.1. Morphological Analysis of Bioscaffold

The fabricated bioscaffold appeared translucent for the A bioscaffold, white-yellowish for the AK bioscaffold, and blackish for the bioscaffold incorporated with carbon. High-resolution 10 kV SEM characterization was performed to examine the surface morphology of the developed bioscaffolds, as shown in Figure 4. Based on Figure 4, the outer surface of the agar bioscaffolds showed a relatively rough surface, whereas the outer surface of the other fabricated bioscaffolds was rougher. In addition, the cross-section image of the bioscaffolds presented interconnected pores with undefined pore structure and size. The interconnected pore allowed for a better exchange of cells’ metabolic waste and nutrients from the growth media [52,53]. The porosity of the bioscaffold was also important, as it influenced the infiltration and development of cells and the integration of host tissue, and it provided a structure for vascularisation [54].

#### 3.2.2. Porosity Analysis of Bioscaffold

The porosity of the bioscaffold was analysed using ImageJ software based on the SEM images of each fabricated bioscaffold, and the results are depicted in Figure 5B. The percentage of the bioscaffold’s porosity was evaluated from five cross-section images of the fabricated bioscaffold in dry conditions. The AK bioscaffold had the highest porosity (36.67 ± 8.1%) as compared to other fabricated bioscaffolds. Moreover, agar, 1HC, 3HC, 1AHC, and 3AHC had average percentages of porosity of 31.35 ± 7.92%, 20.77 ± 5.58%, 24.55 ± 6.33%, 19.37 ± 6.78%, and 27.1 ± 10.24%, respectively. This result indicated that the addition of carbon into the bioscaffold matrix reduced the porosity when compared to the AK bioscaffold. Overall, the series of fabrication methods used in this study successfully produced a porous bioscaffold, which endowed an alternative method when a freeze dryer was not available.

#### 3.2.3. Fourier Transform Infrared Analysis

The diffraction intensity of the bioscaffold, as analysed using FTIR, was shown with peaks in the range of 4000–500 cm^−^¹, as displayed in Figure 6. FTIR spectra of every type of bioscaffold showed characteristics absorption bands at ~3241 cm^−1^, demonstrating the stretching of the hydroxyl (–OH) group and peaks at ~2933 cm^−1^, indicating C–H stretching associated with the ring methine hydrogen bond. Moreover, the stretching of C–O of the hydroxyl group was found at around 1636 cm⁻^1^ [37,55]. In addition, at 1032 cm⁻^1^, the vibration of the pyranose ring C–O–O skeleton was detected [56]. Overall, the peak properties showed no changes between A, AK, and CHAK, indicating minor chemical and structural changes in the constituent molecules of the bioscaffolds. This indicates that the combination of carbonised human hair, agar, and KGM is compatible.

#### 3.2.4. Swelling Ratio Measurement

A bioscaffold’s swelling ability is important for tissue engineering applications. Therefore, the fabricated bioscaffold’s swelling capability was tested and is shown in Figure 7A. All of the fabricated bioscaffolds demonstrated above 1000% of swelling, similar to previous studies [56,57]. The AK bioscaffold exhibited the highest swelling capability (>2000%) among the other fabricated bioscaffolds. Based on Figure 7, the blending of KGM in the bioscaffold enhanced the swelling ability of the bioscaffold. This is because of the high hydroxyl (–OH) groups in the molecular chain of KGM, which offered robust adhesion and mechanical properties [58]. Therefore, the water absorption of the fabricated bioscaffold was enhanced. Moreover, the 3HC bioscaffold showed the lowest swelling capability (<1500%) among the other fabricated bioscaffolds.

In general, the swelling capability of the bioscaffold lessened with the incorporation of carbon, especially when compared to the AK bioscaffold. However, among CHAK, the 3AHC bioscaffold had better swelling capability in comparison to the 3HC bioscaffold, which could be because of the porous structure on the AHC surface, which could take up more liquid. In addition, some of the KGM molecules were displaced with the non-porous carbon within the 3HC bioscaffold, causing lowered HC-based bioscaffold swelling capability. In addition, the fabricated bioscaffold exhibited good water uptake capability (>90%), as shown in Figure 7B.

#### 3.2.5. Degradation Analysis

A bioscaffold’s ability to degrade is essential for enabling the development of the cells’ extracellular matrix. Before the restoration or regeneration of tissue at the implanted site is finished, a good bioscaffold should not start to break down. Therefore, the fabricated bioscaffold degradation rate was evaluated with PBS at room temperature for 28 days and is depicted in Figure 8. Based on Figure 8, the A bioscaffold showed the lowest rate of degradation among the fabricated bioscaffolds after 28 days. In contrast, the AK bioscaffold exhibited the highest degradation capability. According to the graph, the CHAK group had a slightly lower degradation rate compared to the AK bioscaffold. Apart from the A bioscaffold, the overall degradability of the fabricated bioscaffolds reached about 35% after 28 days. In comparison to the A bioscaffold, the inclusion of KGM may weaken the matrix causing the bioscaffold to be more prone to deterioration. Additionally, although it was not noticeably slower, the carbon addition slowed down the produced bioscaffold’s rate of degradation.

Cells will be able to proliferate and differentiate on a bioscaffold with slow degradation before being administered to the intended treatment site. Typically, stem cells need to develop into the appropriate adult cell lineage for longer than 14 days [59]. Therefore, it is crucial to have a robust bioscaffold that gives cells enough time to mature and infiltrate in vitro as well as enough time to build tissue matrices in vivo. As a result, this work effectively manufactured a robust bioscaffold for use in tissue engineering applications based on the degradability evaluation. However, additional degradation research on the produced bioscaffold is necessary, using study parameters that mirror the pH and temperature of the human body.

#### 3.2.6. Tensile Strength Analysis

The bioscaffold’s mechanical strength is essential to ensure that the bioscaffold can withstand the compressive force produced by cells and tissue. In this study, we investigated the fabricated bioscaffold tensile strength using a universal testing machine. The results of the tensile strength test were displayed in the form of tensile strength values (kPa) and are shown in Figure 9. The bioscaffolds were sprayed with PBS to approximately 30% wet states before analysis. Based on Figure 9, the 3HC bioscaffold displayed the highest tensile strength (62.35 ± 29.12 kPa) compared to the other fabricated bioscaffolds, indicating that 3HC is the toughest scaffold. The bioscaffold tensile strength values of A, AK, 1HC, 1AHC, and 3AHC were 54.12 ± 33.53 kPa, 61.25 ± 17.60 kPa, 47.04 ± 19.42 kPa, 42.60 ± 10.69 kPa, and 53.58 ± 20.28 kPa, respectively. Among all, 1AHC exhibited the lowest tensile strength. Overall, the addition of 0.01 g of carbon weakened the tensile strength of the fabricated bioscaffold.

Moreover, the addition of 0.03 g of carbon strengthened the tensile strength of the bioscaffold. One study reported that the concentration or proportion of materials influences the mechanical strength of the bioscaffold [60]. However, there were no significant differences in tensile strength between the fabricated bioscaffold samples analysed. In addition, a study showed that the addition of carbon improves the mechanical properties of the resulting bioscaffold [61]. Their findings are similar to those discovered in our study, particularly at higher concentrations of carbon particles.

#### 3.2.7. Electrical Conductivity Analysis

Electrical conductive materials may influence the activity of stem cells, such as differentiation, migration, and proliferation [62]. Thus, the development of conductive bioscaffolds, which mimic the behaviour of the cell’s natural microenvironment, is rising, especially for cardiac and skeletal muscle tissue engineering applications [62,63]. Hence, the electrical conductivity of the fabricated bioscaffolds was investigated and is shown in Figure 10. Without carbon and KGM, the A bioscaffold initially offered good electrical conductivity. According to a study, agar acts as a precursor to the synthesis of cobalt tungsten powders, which produce electrodes that resemble batteries, demonstrating the good electrical performance of powders made from agar [64]. The ability of pure agar hydrogel to act as a conductive substance is further supported by [65].

Additionally, the electrical conductivity of the bioscaffold was somewhat decreased by the addition of KGM; however, this effect was not particularly significant. In addition, the electrical conductivity of the bioscaffold increased with higher carbon particle concentrations when the electrical conductivity of the CHAK was compared. Significantly, the 3AHC bioscaffold exhibited the highest electrical conductivity (21.14 ± 2.29 µS/m) among the other CHAK. A, AK, 1HC, 3 HC, and 1AHC had electrical conductivities of 20.07 ± 0.57 µS/m, 18.88 ± 2.44 µS/m, 11.83 ± 2.28 µS/m, 16.18 ± 3.01 µS/m, and 15.86 ± 1.89 µS/m, respectively. The 1HC bioscaffold exhibited the lowest electrical conductivity among the fabricated bioscaffolds. According to one study, the existence of pores in activated carbon can function as an ion transfer channel, improving the bioscaffold’s electroconductivity [66].

### 3.3. Cell Culture

#### 3.3.1. Characterisation of WJMSCs

Before biocompatibility analysis, the extracted WJMSCs were characterised in accordance with the standard criteria required for distinguishing MSCs established by the International Society for Cell and Gene Therapy (ISCT) [67]. Therefore, the morphological characteristics of cells, surface markers, and differentiation ability were analysed and are depicted in Figure 11.

##### Morphology of WJMSCs

The morphology of the extracted WJMSCs showed a spindle shape and fibroblastic characteristics similar to other studies, as seen in Figure 11A [68,69].

##### Immunophenotyping of WJMSCs

The surface marker of WJMSCs was analysed and is shown in Figure 11B. The CD90, CD44, CD105, and CD73 expressions of the WJMSCs were 96.96%, 99.16%, 97.81%, and 98.81%, respectively. In addition, the WJMSCs did not express any significant negative lineages expression. These findings were similar to those of other researchers, which showed that the expression of MSC positive markers (CD90, CD44, CD105, and CD73) was above 85% [70,71,72].

##### Differentiation Ability of WJMSCs

The differentiation ability of the extracted WJMSCs was characterised and is shown in Figure 11C. The WJMSCs were stained with anti-mFABP4, anti-hOsteocalcin, and anti-hAggrecan to stain the adipocytes, osteocytes, and chondrocytes, respectively. Under a fluorescent microscope, the WJMSCs showed fluorescent cells indicating that the WJMSCs could differentiate into adipocytes, osteocytes, and chondrocytes. Overall, the extracted WJMSCs adhere to the standard criteria set by the ISCT, indicating that it is a mesenchymal stem cell-like cell.

#### 3.3.2. Biocompatibility Analysis

The bioscaffold’s biocompatibility was investigated by WJMSCs using the MTT assay, and the viability of the cells is shown in Figure 12. After 1 day of incubation, the cell viability of A, AK, 1HC, 3HC, 1AHC, and 3AHC bioscaffold was 77.44%, 78.01%, 117.69%, 77.68%, 103.11%, and 116.52%, respectively. Overall, the cell viability outcomes of fabricated bioscaffolds showed negligible cell toxicity. After 3 days of incubation, the cell viability of A, AK, 1HC, 3HC, 1AHC, and 3AHC bioscaffold was 49.18%, 38.20%, 44.40%, 34.57%, 43.99%, and 47.78%, respectively. All the cell proliferations decreased after 3 days of incubation, which may be due to the leaching of some impurities from the bioscaffold. However, after media change and incubation for 7 days, the growth of cells improved on CHAK (1HC: 109.98%; 3HC: 103.37%; 1AHC: 101.38%, 3AHC: 111.58%) when compared to A (71.48%) and AK (62.72%) bioscaffolds. A study showed similar cell viability outcomes in the existence of hydrogel-containing carbon nanofiber, namely, that the viability of cells was reduced after 72 h, but after 96 h of incubation, the viability of the cells showed enhancement, which they suggested was due to the cells adjusting to the existence of the hydrogel materials [73].

## 4. Conclusions

In this work, the CHAK made of carbon derived from human hair and composite materials consisting of KGM and agar via gelation, rapid freezing, and ethanol immersion processes was successfully fabricated. The fabricated bioscaffold’s micrograph by SEM showed an uneven surface and asymmetrical pore shape. Our findings demonstrated that the addition of carbon reduced the porosity of the bioscaffold. The combination of carbon with agar and KGM is compatible based on the FTIR result because there were no significant changes in wavelength that could be observed between the wavelength of the A, AK, and the bioscaffold incorporated with carbon. Insignificant slight differences among each tensile strength of the bioscaffold were observed, but the 3HC bioscaffold exhibited the highest tensile strength (62.35 ± 29.12 kPa). Moreover, the fabricated CHAK showed relatively slow degradability (<50% after 28 days of incubation), good swelling (>1000%), and excellent water uptake capability (>90%). The 3AHC bioscaffold had the highest electrical conductivity (21.14 ± 2.29 µS/m) among the fabricated bioscaffold. Lastly, the biocompatibility analysis with WJMSCs showed that the CHAK is compatible (>90% cells viability of days 7). In conclusion, this study’s outcomes showed the feasibility of producing carbon from human hair as a material to construct a sustainable carbon-based bioscaffold for tissue engineering applications. However, further study is needed to evaluate its efficacy as a cell culture platform.

## Figures and Tables

**Figure 1 polymers-14-05489-f001:**
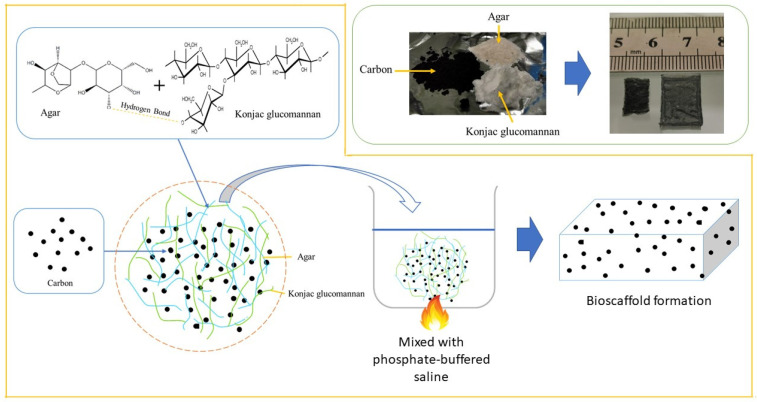
Schematic illustration of the bioscaffold formation.

**Figure 2 polymers-14-05489-f002:**
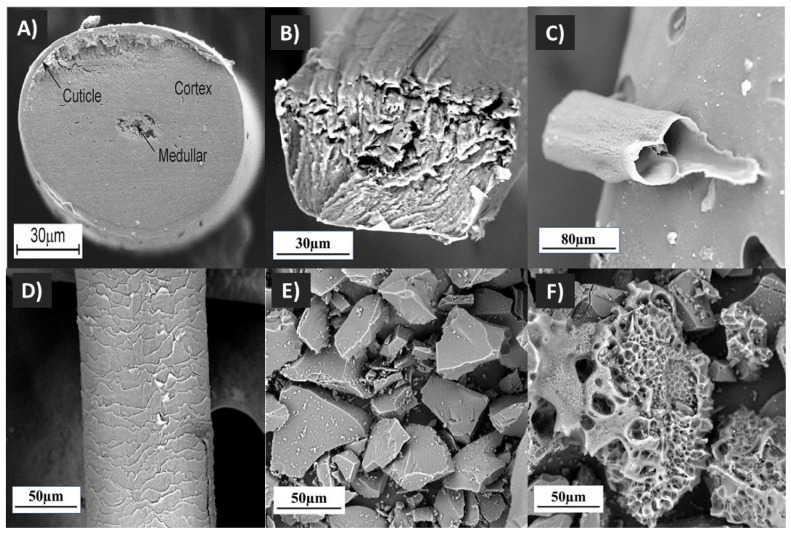
SEM micrographs of raw and carbonised human hair. (**A**) Cross section of human hair adapted from [44]. (**B**) Cross-section of human hair used in this study. (**C**) Carbonised human hair with a hollow microfibre pattern. (**D**) Human hair with cuticle layering around the fibre. (**E**) Non-activated carbon. (**F**) Activated carbon with porous structure.

**Figure 3 polymers-14-05489-f003:**
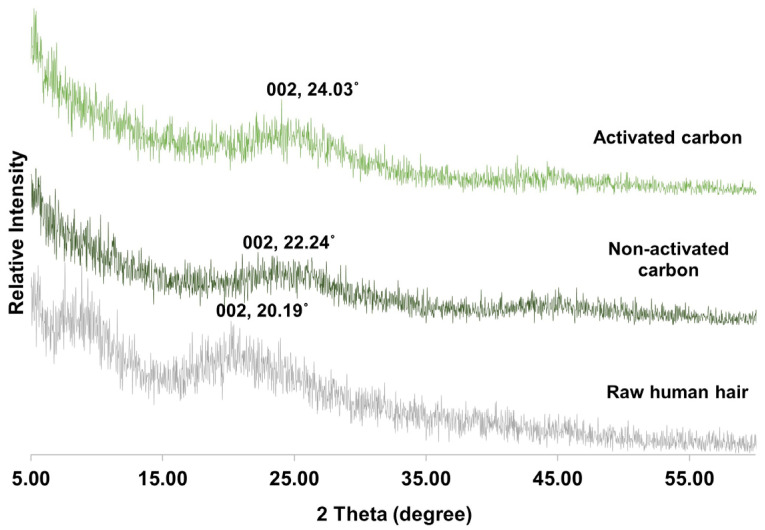
XRD results of raw human hair, non-activated carbon, and activated carbon.

**Figure 4 polymers-14-05489-f004:**
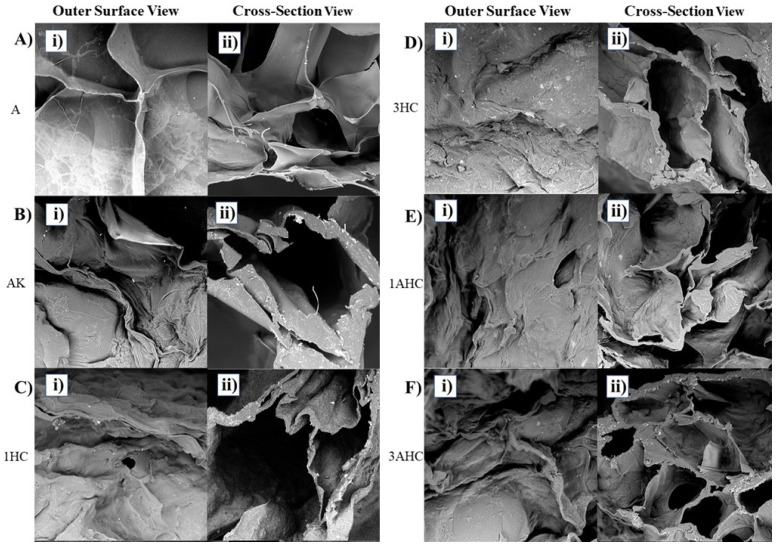
SEM micrographs of the fabricated bioscaffold. Each bioscaffold displayed (**i**) similar outer surface and (**ii**) cross-section features. (**A**): agar; (**B**): agar/KGM; (**C**): agar/KGM/0.01 g HC, (**D**): agar/KGM/0.03 g HC, (**E**): agar/KGM/0.01 g AHC; and (**F**): agar/KGM/0.03 g AHC.

**Figure 5 polymers-14-05489-f005:**
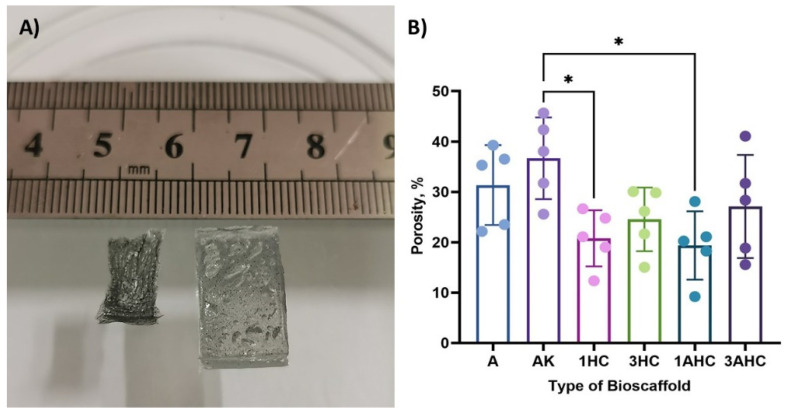
(**A**) Image of dried and rehydrated CHAK. (**B**) The porosity of the fabricated bioscaffolds. The dots on the graph represent individual value of each sample in the group. The statistical significance was considered at a probability of *p* < 0.05 (* *p* ≤ 0.05).

**Figure 6 polymers-14-05489-f006:**
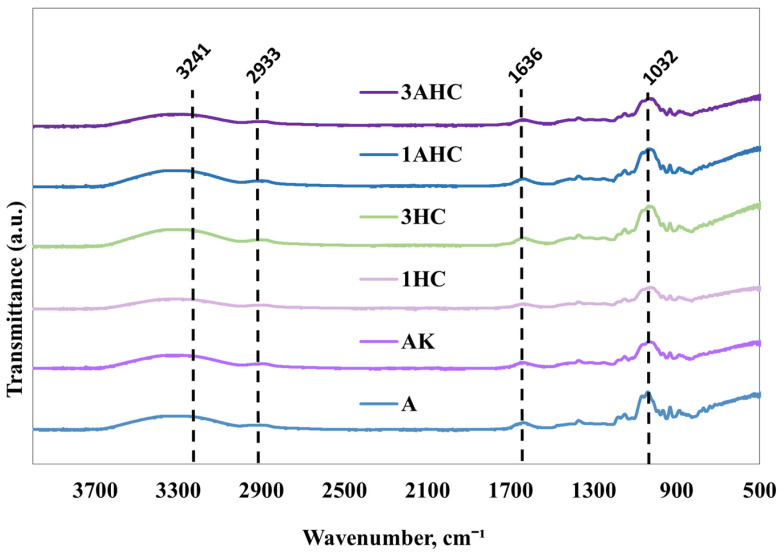
FTIR results of agar, agar/KGM, and CHAK.

**Figure 7 polymers-14-05489-f007:**
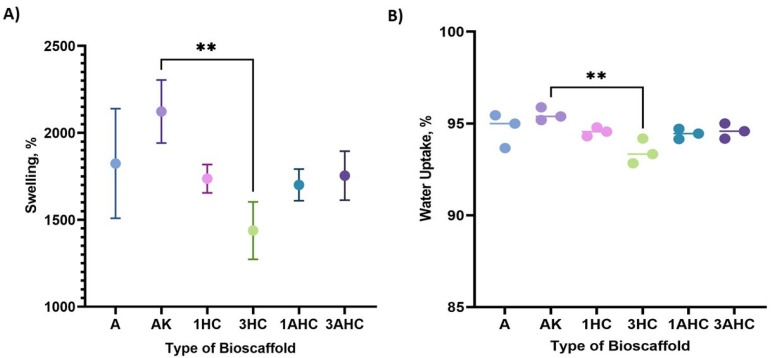
(**A**) Swelling capabilities and (**B**) water uptake of bioscaffold results. The dots on the graph represent individual value of each sample in the group. The statistical significance was considered at a probability of *p* < 0.05 (** *p*≤ 0.01).

**Figure 8 polymers-14-05489-f008:**
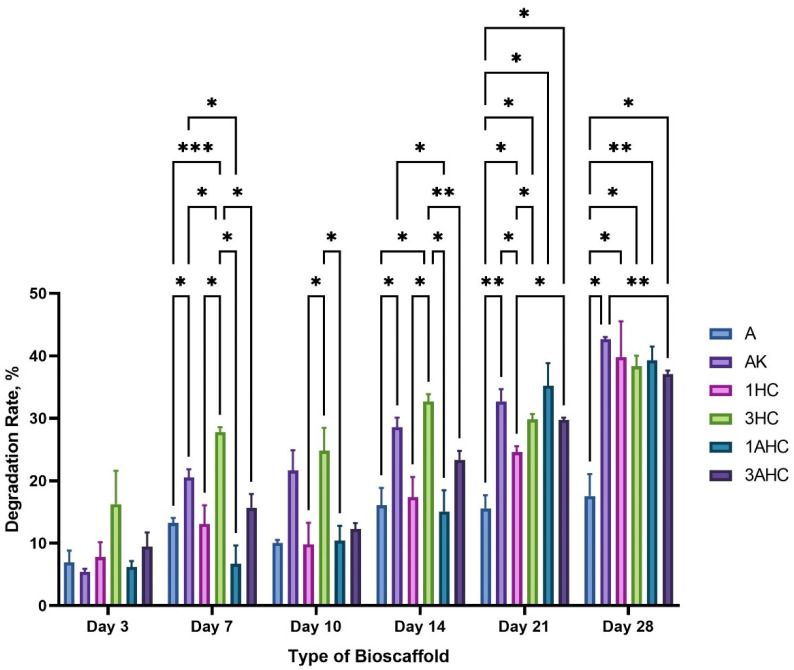
Degradation rate results of bioscaffolds taken on days 3, 7, 10, 14, 21, and 28. The statistical significance was considered at a probability of *p* < 0.05 (* *p* ≤ 0.05, ** *p* ≤ 0.01, *** *p* ≤ 0.001).

**Figure 9 polymers-14-05489-f009:**
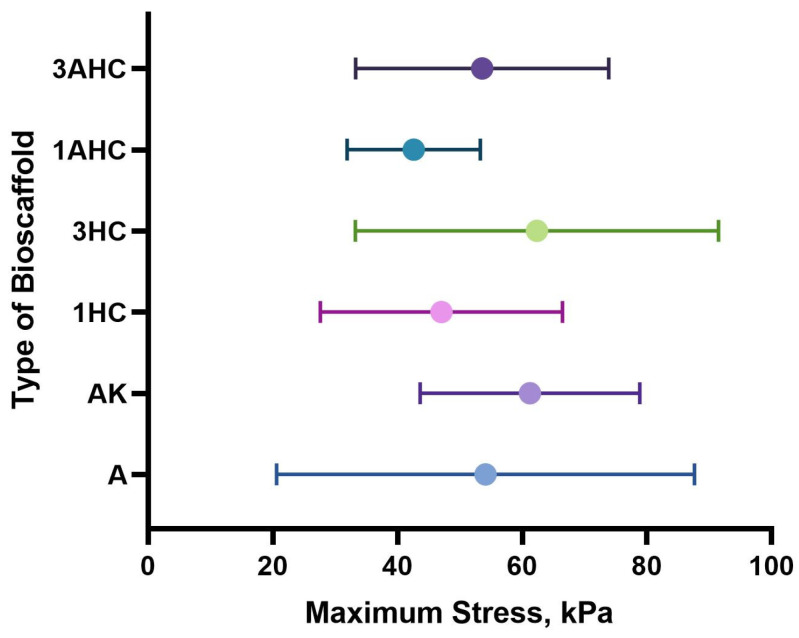
Tensile strength results of bioscaffolds.

**Figure 10 polymers-14-05489-f010:**
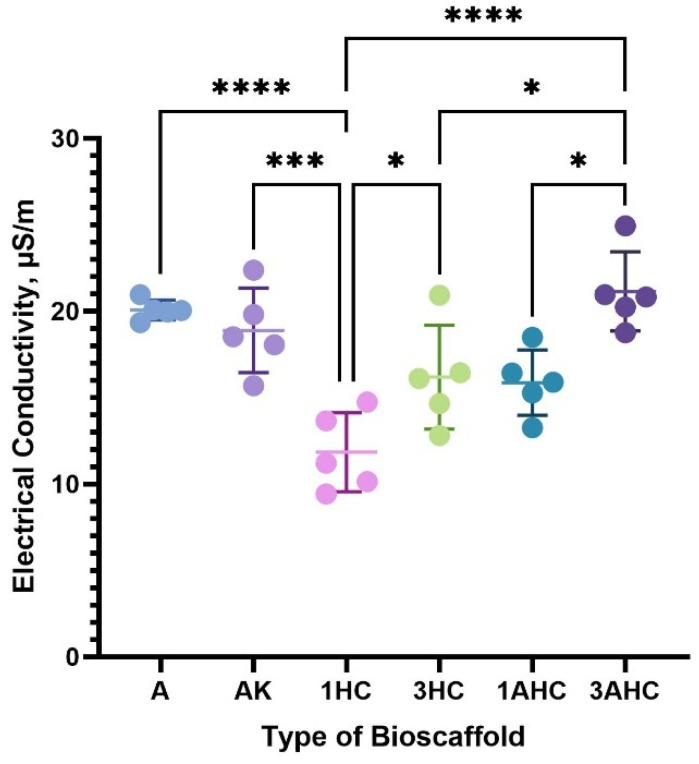
Electrical conductivity results of bioscaffolds. The 3AHC bioscaffold showed the highest electrical conductivity among the fabricated bioscaffolds. The statistical significance was considered at a probability of *p* < 0.05 (* *p* ≤ 0.05, *** *p* ≤ 0.001, **** *p* ≤ 0.0001).

**Figure 11 polymers-14-05489-f011:**
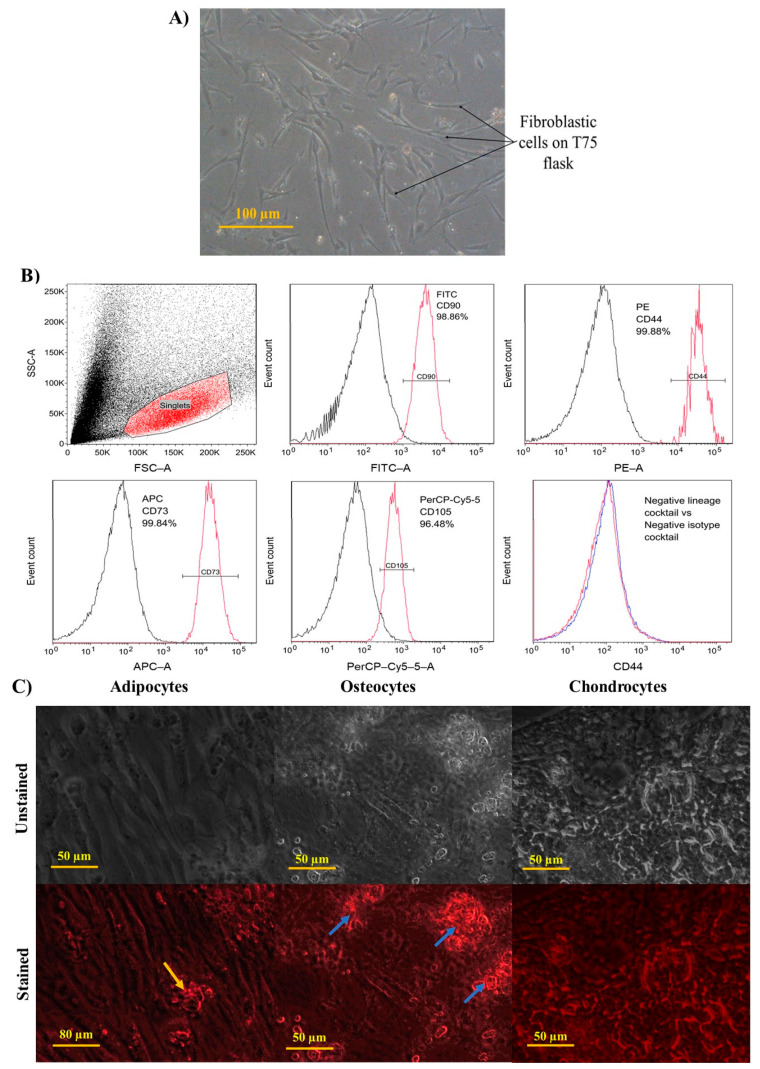
Characterisation of WJMSCs. (**A**) Morphology of WJMSCs showed fibroblastic characteristics. (**B**) Immnunophenotyping of WJMSC. The isolated WJMSCs displayed positive CD90, CD44, CD105, and CD73 expressions (red line). Each marker’s isotype is displayed by a black line. The negative lineage cocktail (red line) against the negative isotype cocktail (black line) showed negative expression of PE fluorescent. (**C**) Differentiation analysis of WJMSCs. Adipocytes, osteocytes, and chondrocytes are depicted in the figure as both unstained and stained cells. The developed cells are indicated by red fluorescence. The lipid droplet is indicated by a orange arrow. Calcium deposits are indicated by blue arrows. The image was taken under a 40× objective lens.

**Figure 12 polymers-14-05489-f012:**
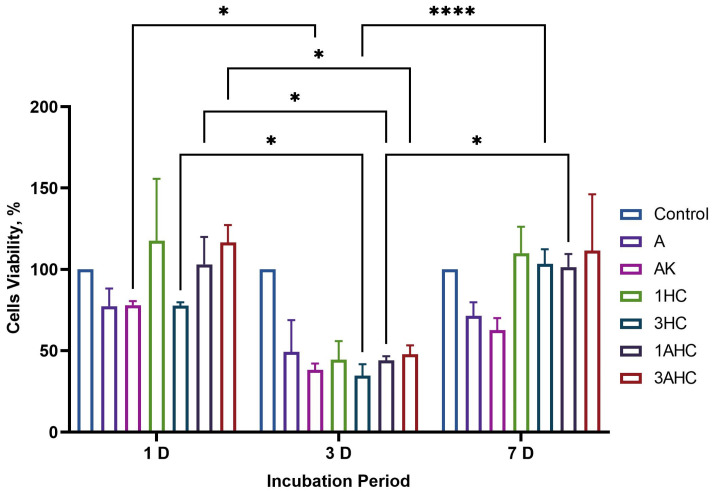
The percentage of cell viability was taken after 1 day, 3 days, and 7 days incubation. The cells’ viability decreased after 3 days, but an improvement in cells’ viability was observed after 7 days of incubation on every fabricated bioscaffold. On day 7, the CHAK bioscaffold showed better cell viability compared to the non-added carbon bioscaffold. Statistical significance was considered at a probability of *p* < 0.05 (* *p* ≤ 0.05, **** *p* ≤ 0.0001).

## Data Availability

Not applicable.

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
