# Peer review of "Carbonised Human Hair Incorporated in Agar/KGM Bioscaffold for Tissue Engineering Application: Fabrication and Characterisation"

_polymers, 2022, doi:10.3390/polym14245489_

Round 1

Reviewer 1 Report

The work is interesting and also have new things to study. I suggest you all to further go ahead in order to explore as per standards set by related organizations. The overall work is good.

Reviewer 2 Report

The authors used carbonized human hair to incorporate with Agar/KGM bioscaffold. The idea is interesting however, it is not very clear if this method or material is better than other reported materials. Also, there are several issues that that authors should be addressed.

(1) The authors should provide a schematic showing the structure and chemical reactions of the developed materials and applications.

(2) The authors should explain why 50 °C was chosen for the degradation test. The authors should use collagenase I for the in vitro degradation test.

(3) It is not clear that how the author incubated cells with the scaffold. Was it on the surface of the scaffold? Why not doing 3D cell culture?

(4) In Figure 6, the authors should use a bar chart because each sample has no relation.

(5) The authors should also provide stress-strain curves for the different bioscaffolds in Figure 8.

(6) Scale bars are missing in Figures 10A and 10C. Moreover, the quality of stained micrographs should be improved. It is difficult to see stained lipid droplets and calcium deposits.

(7) The plots in Figure 11 are confusing, especially the error bars. 

Round 2

Reviewer 2 Report

The authors have addressed the comments and made a major revision. The current version can be accepted.